# Antioxidative NAC-Loaded Silk Nanoparticles with Opening Mucosal Tight Junctions for Nasal Drug Delivery: An In Vitro and In Vivo Study

**DOI:** 10.3390/pharmaceutics14061288

**Published:** 2022-06-17

**Authors:** Tze-Wen Chung, Ting-Ya Wu, Zheng-Yu Siah, Der-Zen Liu

**Affiliations:** 1Biomedical Engineering Research and Development Center, National Yang-Ming Chiao-Tung University, Taipei 112, Taiwan; 2Department of Biomedical Engineering, National Yang-Ming Chiao-Tung University, Taipei 112, Taiwan; wuyaya0317@gmail.com (T.-Y.W.); zyu5771@gmail.com (Z.-Y.S.); 3Biomedical Materials and Tissue Engineering, Taipei Medical University, Taipei 110, Taiwan; tonyliu@tmu.edu.tw

**Keywords:** SF/NAC NP, nasal NAC delivery, anti-oxidative, tight junction, mucosal cells, IVIS study

## Abstract

Using nasal routes to deliver drugs to the brain using multifunctional nanoparticles (NPs) to bypass the blood–brain barrier (BBB) might enhance the delivery efficacy. Anti-oxidative N-Acetyl-L-cysteine (NAC)-loaded silk fibroin (SF/NAC) NPs are produced, characterized and studied as a potential delivery vehicle for NAC delivered to the brain via nasal for both in vitro and in vivo studies. The NPs are not cytotoxic to RPMI 2650 cells, mucosal model cells, at a concentration of 6000 μg/mL. The anti-oxidative activities of SF/NAC NPs are demonstrated by high H_2_O_2_ scavenge capacities of the NPs and shown by mitochondrial superoxide (MitoSOX) immunostaining of human mesenchymal stem cells. Tight junctions in RPMI 2650 cells are opened after 30 min of incubation with SF/NAC NPs, which are demonstrated by measuring the decrease in trans-epithelial electrical resistance (TEER) values and discreteness in ZO-1 stains. The cellular uptake of SF/NAC NPs by RPMI 2650 cells is significantly greater than that for SF NPs and increased with increasing incubation time. In an in vivo imaging study (IVIS) using rats shows that the amount of NAC that is delivered to the brain by SF/NAC NPs increased by 1.40–2.60 times and NAC is retained longer in the nasal cavity than NAC solutions in a 2-h study.

## 1. Introduction

Silk fibroin (SF) is a natural macromolecule that is characterized by hydrophobic blocks of the repeated amino acid sequence (Gly–Ser–Gly–Ala–Gly–Ala)_n_, which contains several chemically highly active amino acids, such as tyrosine [1]. Sericin-free SF has a low immune-inflammatory response and excellent biological response [2], so it is a suitable biomaterial for engineering various tissues, such as cardiac patches, hydrogels and scaffolds, refs. [3,4,5] and drug delivery in varying forms (e.g., NPs or hydrogels) or with multifunctional properties such as photothermal responses [6,7,8]. The SF polymer also contains several hydrophilic and hydrophobic segments, and the crystallinity of the polymer is easily modulated, so it is a suitable biomaterial for drug delivery vehicles [9,10]. Drug delivery applications for SF NPs or SF-based NPs are largely focused on cancer or other therapies [8,9,10]. Applications for SF-based NPs for nasal drug delivery have not been reported. This study hypothesizes that SF NPs with a loaded drug, NAC, might be adsorbed onto the nasal mucosa during nasal drug delivery.

Drug delivery routes are generally categorized as parenteral, such as intravenous and intramuscular administrations, or as mucosal routes, such as oral and intranasal delivery. Nasal drug delivery is an alternative method to access the brain via the nasal cavity because the specific anatomic structure of the nasal cavity gives a direct connection between the extensions of axons or nerve filaments from the olfactory bulb in the limbic region of the brain to the upper posterior segments of the nose, so the blood-brain barrier (BBB) is avoided by parenteral drug delivery [11,12].

Another pathway for nasal drug delivery to the brain that does not involve passing through the BBB uses cranial nerves, such as the trigeminal nerve system [13]. Nasal mucus is a weak acid with a pH of 5.5–6.5 with a negative electric charge because it contains a large number of OH^−^ groups and oligosaccharide chains in the regions of the nasal epithelium [14,15,16]. A disadvantage of nasal drug delivery is fast mucociliary clearance of the nasal cavity (e.g., less than 30 min), which clears mucus and substances that adhere to the mucosa, such as pathogens, particles and administered substances, and drains them into the nasopharynx. Therefore, increasing the retention time for drugs or drug-loaded nanoparticles in the nasal cavity is a strategy to enhance nasal drug delivery.

To increase the effectiveness of nasal drug delivery, increasing the residence time in the nasal cavity is achieved using delivery by hydrogels, which increases the viscosity of the delivering media [16,17]. An absorption enhancer such as chitosan (CS) is used to increase the permeability of the nasal epithelium [17,18]. Early reports show that CS in the form of nanoparticles or hydrogels is a polymer that increases the permeation of drugs by opening up the tight junctions in the nasal epithelia and increasing the rate of absorption for drugs such as insulin [16,17,18].

To determine whether the opening of epithelial layers in tight junctions, a protein complex which mainly consists of Occluding, Claudin, E-cadherin, ZO-1, Jam and other substances [19] is used and the impedance values for trans-epithelial electrical resistance (TEER) are measured and ZO-1 stains are carried out [20,21]. For nasal drug delivery, there is no study on whether drug-loaded SF NPs (e.g., SF/NAC NPs) could open the tight junctions of nasal epithelia in the nasal cavity as those reports for CS solutions or CS-based NPs or hydrogels [16,17,18]. This study produced drug-loaded SF nanoparticles and determines their efficacy for nasal drug delivery.

N-Acetyl-L-cysteine (NAC) is a membrane-permeable cysteine precursor that is a rate-limiting substrate for GSH biosynthesis and a free radical scavenger or an antioxidant that is involved in the intracellular and extracellular detoxification of reactive oxygen species in the brain [22]. NAC increases blood glutathione redox ratios in subjects with Parkinson’s. There is an increase in brain glutathione concentrations for all study subjects after one intravenous dose of NAC, but the study involves small samples. [22]. However, the clinical application of NAC is limited by its low bioavailability and short half-life. To address those limitations, NAC conjugated polymer is used to produce NPs and is more effective than free NAC in decreasing the effect of LPS on microglia and an in vitro study shows that neurons are protected [23]. A recent study shows that NAC released from cysteine (NAC)-loaded poly (lactic-co-glycolic acid) (PLGA) electrospinning scaffolds improves the in vitro viability and proliferation of rat pheochromocytoma (PC12) and human glioblastoma multiform (A172) cells, so this can be used to regenerate neural tissue after traumatic brain injury (TBI) [24]. In this study, SF/NAC NPs were produced, characterized and examined for their potential for in vitro NAC delivery in a model of mucus cells and in vivo nasal NAC delivery in a rat model.

The techniques to produce drug-loaded SF-based NPs, such as de-solvation/coacervation, supercritical fluid and microemulsion methods, have been documented recently [9]. For instance, a water-miscible organic solvent containing a hydrophobic drug (e.g., paclitaxel) was slowly injected into an SF solution to produce paclitaxel-loaded SF nanoparticles for an antitumor study [8]. Using a similar process, we slowly injected NAC, a hydrophilic drug, dissolved in an acetone solution into an SF solution to produce SF/NAC NPs for the evaluation of nasal NAC delivery. The NPs were characterized in terms of particle size, cytotoxicity to mucosal cells, RPMI 2650 and anti-oxidative activity to human bone marrow mesenchymal stem cells (hBMSC). To determine the suitability of SF/NAC NPs for nasal delivery to the brain, RPMI 2650 cells were used as model mucosal cells. The values of TEER and ZO-1 stains were then measured to determine whether tight junctions were opened and to determine whether there was a cellular uptake of the NPs by cells after they were incubated with SF/NAC NPs. A preliminary rat study to determine the efficacy of nasal delivery of NAC to the brain by intranasal administration of fluorescent SF/NAC NPs used an IVIS study. In vitro and in vivo studies were used to show whether SF/NAC NPs could be used for nasal delivery to the brain.

## 2. Experimental Methods

### 2.1. Preparing SF Solutions

Silk cocoons were provided by a silk center, MDARES (Miaoli Agricultural Research and Extension Station, Council of Agriculture, Executive Yuan, Miaoli, Taiwan). A solution of SF (MW~185 kDa) was prepared as described in a previous study by the authors [7,25] and involved degumming in 9.3 M LiBr and removal of Li^+^ using dialysis with deionized (DI) water. In brief, silk cocoons were boiled in 0.02 M Na_2_CO_3_ for 90 min and then rinsed thoroughly in DI water to extract the glue-like sericin proteins from silk fibroin. The extracted SFs were dissolved in 9.3 M LiBr solution at 75 °C for 1 h and then dialyzed in DI water using a dialysis membrane (MWCO 6000) (Spectra/Por1, Repligen Co., Waltham, MA, USA) at room temperature for 48 h to remove salts. The final concentration of the SF aqueous solution was 20% (*w*/*v*) [7,25].

### 2.2. Fabrications of SF/NAC NPs

Solvent dissipation or simple coacervation process was used to produce SF/NAC NPs [8,9]. In brief, 45 mg NAC (N-Acetyl-L-cysteine, Sigma-Aldrich, St. Louis, MO, USA) was dissolved in acetone solution, slowly injected dropwise using an infusion pump into 15 mL of 1.5% SF solution containing 0.1% PVA (Tokyo Chemical Industry, TCI, Tokyo, Japan) in gently stirring. SF/NAC NPs were produced very fast and suspended in SF solutions. The suspensions containing the NPs were centrifuged at 7500× *g* for around 15 min several times to remove free NAC in membrane tubes with a cut-off MW. of 3 kDa, and the concentrated SF/NAC NPs suspensions were quantified for further experiments.

### 2.3. Characteristics of SF and SF/NAC NPs

The particle size and zeta potentials of the SF/NAC NPs were measured using a dynamic light scattering particle size/zeta potential analyzer (Particulate Systems NanoPlus, Norcross, GA, USA). The SF/NAC NPs were suspended at a concentration of 0.05 mg/mL in DI water for further measurements [7,26]. The morphological images of SF/NAC NPs were carried out by a transmission electron microscopy (TEM) with an accelerating voltage of 100 kV (JEOL JEM-2000EX II, Tokyo, Japan) after they were stained with 2% (*w*/*v*) phosphotungstic acid and placed on copper grids [7,26].

### 2.4. ATR-FTIR Spectra to Analyze the Functional Groups of SF and SF/NAC NPs

ATR-FTIR spectra were performed using an ATR-FTIR (Attenuated Total Reflectance-Fourier Transform infrared) spectrometer (IRAffnity-1, Shimadzu Co., Kyoto, Japan) to examine the functional groups of SF, NAC and SF/NAC NPs. Detailed procedures for the measurements could be referred elsewhere [7,26]. For this study, samples were scanned from 600 to 4000 cm^−1^ at a resolution of 4 cm^−1^ and the spectra were analyzed using the built-in standard software package [7,26].

### 2.5. H_2_O_2_ Scavenging Abilities of SF and SF/NAC NPs Using a MitoSOX Immunostaining

For this study, hBMSC cells were cultivated according to the protocols described in an early report of this Lab. [27]. In brief, 3 × 10^5^ hBMSC (PT-2501, Lonza Group Ltd., Basel, Switzerland) were cultivated in 10 cm culture dishes in a medium (MSCGM, PT-4105, Lonza Group Ltd., Switzerland) which was refreshed every 2–3 days. After the cells reached confluence, they were detached by trypsin/EDTA (CC-3232, Lonza Group Ltd., Switzerland) at 37 °C for further cell passaging or experiments. The viability of hBMSC was routinely monitored by staining with Live/Dead or using a viability/cytotoxicity kit (Invitrogen Corp., Carlsband, CA, USA) according to the manufacturer’s instructions. HBMSCs with a passage of 6~8 were used for examining H_2_O_2_ scavenging abilities for SF and SF/NAC NPs.

To determine an adequate amount of H_2_O_2_ for the scavenging abilities of those NPs, different concentrations of H_2_O_2_ (0, 5, 10 and 15 μg/mL) were added to 96-well culture plates with a cell density of 8 × 10^3^ hBMSC/well to study the cell viability using an MTT assay [7,25,26]. After 24 h cultivations, the concentration of H_2_O_2_ that resulted in large decreases in viabilities of hBMSC was chosen for further studying H_2_O_2_ scavenging abilities for SF and SF/NAC NPs [7,26].

For studying H_2_O_2_ scavenging abilities of NPs consisting of different biomaterials, the protocols were described elsewhere [7,26]. In this study, H_2_O_2_ (e.g., 10 µg/mL) was first mixed with SF NPs and SF/NAC NPs (e.g., H_2_O_2_/SF and H_2_O_2_/SF/NAC NPs) at 25 °C for 24 h. HBMSC cells were then incubated with fresh medium (control), H_2_O_2_ (10 µg/mL), and H_2_O_2_/SF/NAC NPs suspensions for 3 h, respectively [26]. The hBMSC cells were then washed with PBS, and a MitoSOX Assay Kit (Abcam, Cambridgeshire, Cambridge, England) was employed to examine intracellular ROS of mitochondria (MitoSOX) of the cells which were labeled by red fluorescence [7,26]. DAPI staining for nuclei (e.g., blue) and MitoSOX immunostaining for ROS of mitochondria (e.g., red fluorescence) of hBMSC cells were observed using a fluorescence microscope (Olympus IX71, Tokyo, Japan) which were reported in detail in early report [7,26].

### 2.6. Cytotoxicity of SF and SF/NAC NPs

L929 cells were purchased from Lonza Corp. (Lonza Group Ltd., Switzerland). The detailed procedures for proliferation and evaluating cytotoxicity of L929 cells for this study are described elsewhere [3,25,26]. According to the guidelines of International Standard Organization (ISO) 10993-5 for evaluating the cytotoxicity of cells for the SF and SF/NAC NPs, the viability of L929 cells was measured using an MTT assay (CGD1, Sigma, USA) after they were incubated with different concentrations of the NPs (e.g., 0~6000 μg/mL) for 24 h [7,26]. In brief, MTT agents were incubated with L929 cells at a concentration of 1.0 × 10^4^ cell/well for 4h in a 96-well plate in the dark and formazan crystals that formed in cells were dissolved in 100 µL of DMSO. The absorbance of the formazan for these samples was measured at a wavelength of 570 nm to determine the cell viability of L929. The results of the cell viability were normalized to calculate the cytotoxicity of different concentrations of SF and SF/NAC NPs [7,26].

### 2.7. Encapsulation and Loading Efficiencies of SF/NAC NPs

Different concentrations of NAC solutions were calibrated by Ellman’s method (DTNB) (5,5′-dithiobis-(2-nitrobenzoic acid) or DTNB) which was dissolved in 0.1 M sodium phosphate buffer (pH 8.0) at 0.15 mg/mL [28]. Briefly, DTNB solutions were mixed with different concentrations of NAC (e.g., 25 μg/mL) at a volume ratio of 1:5, and those mixtures were reacted in a dark for an appropriate time. The absorbance values of the mixtures at a wavelength of 412 nm were determined using an UV/VIS spectrophotometer (UV-2600, Thermo Multiskan GO Microplate Spectrophotometer, Waltham, MA, USA). To determine the amounts of NAC in SF/NAC NPs, 1 mg/mL NPs was dispersed in 95% ethanol for 4 h, diluted by sodium phosphate buffer, and the suspension was centrifuged at 15,000× *g* for about 20 min [7,26]. The concentration of NAC in the supernatant of the suspension was determined by measuring the absorbance at wavelength of 412 nm using the spectrophotometer [28]. The encapsulation and loading efficiencies of NAC (%) were calculated according to the definitions stated elsewhere (e.g., for loading efficiency, the amount of NAC encapsulated in 1mg of SF/NAC NPs) [7,26].

### 2.8. Determining NAC Releases for SF/NAC NPs

To determine the cumulative NAC release for SF/NAC NPs, 10 mg/mL of the SF/NAC NPs in PBS solution was suspended in a dialysis membrane bag with a cut-off MW. of 3500 Da, which was immersed and well stirred in 10 mL PBS (as a dissolution medium) at 37 °C [7,26]. The NAC released in the medium was pipetted out (e.g., 100 μL) at a specific time to examine the concentration of NAC by measuring the absorbance at a wavelength of 412 nm using the spectrophotometer [28]. Fresh dissolution medium was added into the dissolution beaker.

### 2.9. TEER Measurements for Human Nasal RPMI2650 Cells

Human nasal RPMI 2650 cells were purchased from BCRC (Bioresource Collection and Research Center, Hsin-Chu, Taiwan). To cultivate the cells, 7 ${\times \;10}^{6}$ cells were seeded on 10-cm tissue culture dishes (Corning Costar Corp., Corning, NY, USA) in a cultural medium (90% MEM (Corning Corp., New York, NY, USA) with 2 mM L-glutamine (Sigma, St. Louis, MO, USA), 10% fetal bovine serum (FBS, Life technologies-Gibco, Livingstone, NJ, USA) and others) in accordance with the BCRC guidelines. The medium was renewed for every 2–3 days. When cells reached confluence, they were detached using 0.25% trypsin/EDTA (Life Technologies-Gibco Corp., USA) at 37 °C for further passaging or experiments for cell studies.

For the TEER measurements for RPMI 2650 cells, the cells were proliferated as a method that is similar to that for other studies [29,30,31] with minor modifications. RPMI 2650 cells at a density of 5 × 10^4^ cells/0.3 cm^2^ were seeded on a 24-well plate of trans-well inserts (Falcon, New York, NY, USA), using a membrane with 0.4 μm pores. The cells were cultivated in 10% serum-containing medium and continuously proliferated in an air-liquid interface for about 2–3 weeks to yield RPMI cell layers [29,30,31]. The TEER studies were measured the electrical resistance using an EVOM2 Volt-ohm meter with an STX3 electrode (World Precision Instruments, Sarasota, FL, USA) in accordance with the manufacturer’s instructions. The TEER values measured in RPMI 2650 cell layers in this study were about 110~120 Ω·cm^2^ which were similar to the values of human nasal mucosa reported by others [30,31].

To determine the effects of SF/NAC NPs and chitooligosaccharide (COS) on opening the tight junctions in RPMI 2650 cell layers [32], 0.3 mL of medium containing different concentrations of SF/NAC NPs (e.g., 30, 100, 300 µg/mL) or COS (e.g., 0.1, 0.5 and 1%) were added into the cell seeding chambers (World Precision Instruments, Sarasota, FL, USA). The TEER values for evaluating the effects of opening the tight junctions in RPMI 2650 cell layers by the NPs or COS were determined by measuring the electrical resistance using an STX3 electrode and an EVOM2 Volt-ohmmeter (World Precision Instruments, Sarasota, FL, USA) in accordance with the manufacturer’s instructions. The TEER values for these studies were continuously monitored every 30 min in 2 h. For further study, the opening status of tight junctions in RPMI 2650 cell layers after SF/NAC NPs or COS suspensions on the cell layers were washed by PBS solutions at 2 h, and the TEER values for the cell layers incubated in fresh mediums were continuously measured at 4, 6 and 24 h, respectively.

### 2.10. ZO-1 Staining for Examining the Opening Tight Junctions in RPMI 2650 Cells

To determine the effect of SF/NAC NPs (300 µg/mL) or COS (0.5%) solutions on junctional proteins (ZO-1) in RPMI 2650 cells using immunochemical stain [20] after they were incubated for 2 h, the supernatants were removed and the cells were washed by phosphate-buffered saline (PBS) several times and fixed using 0.25% glutaraldehyde for 10–20 min. The cells were further blocked by 5 % bovine serum albumin (BSA) in PBS, incubated with anti-ZO-1 (Invitrogen, Carlsband, CA, USA) overnight at 4 °C, Alexa flour 488 goat anti-mouse IgG (Sigma-Aldrich, USA) for 1 h at 37 °C, and stained with 4′,6-Diamidino-2-Phenylindole (DAPI) for the cell nuclei of the cells [27]. The images for ZO-1 and DAPI stains of the RPMI 2650 cells were observed and taken using a confocal laser scanning microscope (LCSM-ZEISS LSM 880, Zeiss, Oberkochen, Germany).

### 2.11. Rats Studies for Nasal Delivery of FITC-NAC to Brain by Administrations of SF/FITC-NAC NPs into Their Nasal Cavities

For carrying out the IVIS study, the experimental protocols for labeling fluorescein isothiocyanate (FITC) (Sigma-Alderich, St. Louis, MO, USA) to proteins or NAC which contains amine and other functional groups (e.g., -SH groups) were according to manufacture guidelines with few modifications [26]. Briefly, for preparing fluorescein isothiocyanate (FITC) grafted NAC (FITC-NAC), 0.1% fluorescein isothiocyanate (FITC) solution was first dissolved in dimethyl sulfoxide (DMSO)/water at 4 °C for 24 h. NAC was grafted by FITC (e.g., the ratio of NAC to FITC was 100:1) in a DSMO/water solution for about 6h at 25 °C [26]. The non-FITC grafted NAC in the aforementioned solutions was nearly removed by centrifugation at 2000× *g* for 10 min several times using a membrane tube with a cut-off MW. In total, 3 kd (Amicon^®^ Ultra filters, Burlington, MA, USA). The green fluorescent FITC-NAC solutions were routinely examined by irradiating or exciting the samples using the wavelength of 487 nm. To prepare SF/FITC-NAC NPs for nasal delivery of FITC-NAC, the procedures were about the same as those described in Section 2.2 except that FITC-NAC instead of NAC dissolved in acetone was injected into an SF solution to produce SF/FITC-NAC NPs which were gently washed by ethanol to remove FITC residues [26] for rat studies.

Male Sprague Dawley rats (300~350 g) were supplied by BioLasco Co., Ltd. (HsinChu, Taiwan). All in vivo studies were conducted in accordance with the guidelines of the Institutional Animal Care and Use Committee (IACUC) and approved by National Yang-Ming University [7]. The rats were anesthetized with isoflurane before nasal NAC was delivered using a micropipette [33]. For using IVIS study to determine the quantity of FITC-NAC or SF/FITC-NAC NPs that remained in nasal or reached brain regions of rats, 50 uL of FITC-NAC or SF/FITC-NAC NPs (300 µg/mL) in PBS was administrated into the nasal cavity of rats, and fluorescent images of the regions of interest (ROI) were captured at specific times using an optical instrument (Biospace Lab Optima, Nesles-la-vallee, France) for quantifications.

### 2.12. Statistical Analysis

All calculations used SigmaStat statistical software (Jandel Science, San Rafael, CA, USA) [7,25,26]. Statistical significance for the Student t-test corresponds to a confidence level of 95%. Data are presented as mean ± SD for at least three measurements. Differences are deemed to be statistically significant if p < 0.05.

## 3. Results and Discussion

### 3.1. Producing SF/NAC NPs and the Characteristics of the NPs

A previous study used a solvent dissipation process to produce a hydrophobic drug or paclitaxel-loaded SF NPs has been reported [8]. Using a similar concept as Wu et al. [8], we slowly injected NAC, a water-soluble drug, dissolved in acetone solutions, into an SF solution to produce SF/NAC NPs. Since NAC is a hydrophilic drug with functional groups (e.g., carboxyl groups), it would interact with functional groups (e.g., Amide I, presented in ATR-FTIR spectra in Section 3.2) of SF, and NAC would mainly be located at hydrophilic segments of SF, which were different from the locations of paclitaxel within the SF NPs [8]. The particle size and zeta potential for SF and SF/NAC NPs were determined by a DLS and the results are shown in Figure 1A. The spherical morphology of NPs for different magnitudes (e.g., 3 KX and 20 KX) is shown in Figure 1B. SF/NAC NPs (e.g., 227.8 ± 8.8 nm, n = 4) are significantly larger than SF samples (e.g., 174.3 ± 2.5 nm, n = 4). These results were consistent with morphological TEM images showing the size of NPs (e.g., 174.7 nm for SF NP) (Figure 1B). Moreover, the zeta potential for SF/NAC NPs changed from a negative charge for SF NPs to a positive charge (e.g., 6.4 ± 0.9 mV, n = 4) (Figure 1A) because the positive charges of NAC [34] were encapsulated in the SF/NAC NPs. Therefore, the particle size increased and the charges of SF/NAC NPs became positive, in contrast to SF NPs. This probably occurred because NAC was encapsulated into SF NPs.

### 3.2. A Analysis of ATR-FTIR Spectra for Examining Functions Groups SF/NAC NPs

The functional groups of NAC and SF in the SF/NACNPs were determined using the ATR-FTIR spectra. The functional groups and the change in the characteristic peaks were determined by the decrease in the transmittance of the peaks (Figure 2). The characteristic peaks for the functional groups of NAC at 2546 cm^−1^ for the SH group and 1718 cm^−1^ C=O for carboxyl groups were consistent with the results of other studies [35]. The characteristic peaks in the spectra for the functional groups of SF were Amide I, Amide II and Amide III (e.g., 1237 cm^−1^) and were the same as those for SF NPs, which was in agreement with earlier studies [7]. The shift in the peak for Amide I for SF NPs from 1668 cm^−1^ to 1650 cm^−1^ for SF/NAC NPs might be due to chemical interactions between carboxyl groups (-COOH) in NAC and Amide I in SF. The shifts in the peak for Amide II in SF NPs from 1537 cm^−1^ to 1524 cm^−1^ for SF/NAC NPs were possibly due to the presence of Amide II in NAC (Figure 2). There was a minor peak at 2330 cm^−1^ for SF/NAC NPs, which was attributed to the SH groups in NAC, as reported by other studies [36].

### 3.3. Cytotoxicity Examinations for SF and SF/NAC NPs

Cytotoxicity studies of SF and SF/NAC NPs were performed according to the standards of ISO10993-5. Using the cell viability (%) for L929 fibroblasts that was shown in Figure 3, the values for SF and SF/NAC NPs groups at concentrations of less than 6000 μg/mL are similar and are significantly greater than 75% (*p* < 0.01 for all tested concentrations, n = 3, respectively). According to the standards of ISO10993-5, SF and SF/NAC NPs in concentrations of 0~6000 μg/mL were biocompatible. The cytotoxic concentrations for SF and SF/NAC NPs for this study were better than those for SF/dopamine NPs [7] and similar to those for other SF-based biomaterials [37].

### 3.4. Anti-Oxidative Abilities for SF/NAC NPs by Determining the hBMSC Viability

The cytotoxicity studies for SF and SF/NAC NPs gave similar results (Figure 3) and NAC was clinically used as an anti-oxidative drug, so the anti-oxidative effect of SF/NAC NPs was verified. The anti-oxidative ability of SF/NAC NPs was measured by calculating the viability of hBMSC and examining the MitoSOX red stains for cells after they were attacked by H_2_O_2_ at a specific concentration [26]. To determine the minimum concentration of H_2_O_2_ that was required to attack and reduced the viability of hBMSC cells, the cells were incubated with different concentrations of H_2_O_2_ and the results were shown in Figure 4A. In total, 10 μg/mL of H_2_O_2_ reduced the viability of hBMSC cells by 33% and 15 μg/mL of H_2_O_2_ resulted in about 100% death of the cells. Therefore, this study uses 10 μg/mL H_2_O_2_ to determine the anti-oxidative ability of hBMSC cells for different concentrations of SF/NAC NPs (Figure 4B).

Figure 4B showed the effects of different concentrations of SF/NAC NPs on the viability of hBMSC cells that were attacked by 10 μg/mL H_2_O_2_ (n = 3). The results showed that 50 μg/mL or higher concentrations of SF/NAC NPs ensured at least 90% viability for hBMSC cells that were attacked by H_2_O_2_ (Figure 4B). A MitoSOX red stain, which is a mitochondrial superoxide indicator, was used as an assay to determine the damage to the mitochondria of cells that are attacked by free radicals [26]. The results in Figure 4B showed that there was anti-oxidative ability at a concentration of 100 μg/mL SF/NAC NPs could effectively scavenge H_2_O_2_ to prevent damage to MSC cells from H_2_O_2_ using the MitoSOX red stains for the cells (Figure 4C). Few cells were stained red, and the number is almost the same as that for the control group (fresh medium) (Figure 4C). H_2_O_2_ was incubated with the NPs for 24 h prior to incubating with the cells [26], so the results in Figs.4B and 4C showed that with the release of NAC, which is an antioxidant, SF/NAC NPs scavenged H_2_O_2_ to prevent the damage of hMSC cells from being attacking by H_2_O_2_.

### 3.5. Paracellular Transport: Opening Tight Junctions in RPMI 2650 Cells Using SF/NAC NPs by TEER Examinations

Paracellular transport is one of the pathways for nasal drug delivery. The drug passes through different cell junctions in the nasal epithelia, which have tight junctions. A junction of 4.0~8.0 A in diameter is impermeable to drugs with large molecules, so there is low membrane transport for nasal drug delivery besides mucociliary clearance in the nasal cavity. If these tight junctions are not opened, the permeability of the epithelia of the nasal cavity is low and the drug hardly passes, so enhancers or chitosan-based hydrogels or NPs are usually used to increase the rate at which drugs such as insulin are absorbed [16,17,18,19,20]. Trans-epithelial electrical resistance (TEER) is a non-invasive technique that quantitatively measures the impedance across an epithelial monolayer or tissue to show the integrity of tight junctions in the layer in cell culture models.

TEER measurements were used as indicators to determine the transport or interaction of drugs or chemicals through/or with the epithelial monolayer, such as nasal drug delivery by chitosan NPs [18,20]. TEER tests were performed to determine the tight junction dynamics for RPMI 2650 cells for nasal NAC delivery via SF/NAC NPs. RPMI 2650 cells are widely used as an in vitro cell model for the evaluation of intranasal drug delivery of drugs [30,31]. It has been reported that the TEER value for RPMI 2650 cells in a confluent monolayer was 45 ± 6 Ω cm^2^ [30] which was approximate to the value for human nasal epithelial monolayer (75–180 Ω cm^2^) [31]. The TEER value for this study for RPMI 2650 cells in a confluent monolayer was 55.1 ± 3.9 Ω cm^2^, which is consistent with values for other studies [30,31]. It is acknowledged that the greater the decrease in the TEER value, the more opening tight junctions in the epithelial cell layer (RPMI 2650 cells in this study).

Figure 5A showed the dynamic decreases in the TEER values for different concentrations of SF/NAC NPs and chitooligosaccharide (COS), oligomers of chitosan with similar biological properties as those of chitosan [16,32], such as opening the tight junctions in the mucus cell layer that was used for comparisons with the NPs for RPMI 2650 cells that were incubated for 2 h. For the SF/NAC NPs group (Figure 5A(A-1)), the TEER value decreased to 81.7 ± 1.3%, 79.6 ± 1.1% and 67.1 ± 3.0%, (n = 3), as the concentrations of NPs incubated with RPMI 2650 cells increased (e.g., 30, 100 and 300 μg/mL, respectively) for 0.5h, so high concentrations of SF/NAC NPs might open more tight junctions in the cells. However, there were smaller decreases in the TEER values for the NPs group than for the COS group (47.3 ± 3.5%, 34.7 ± 2.6% and 30.0 ± 1.5% for 0.1, 0.5 and 1% COS, (n = 3), respectively.

The decrease in the TEER value for each group was similar to those for 0.5 h to 2 h (Figure 5A). The difference in TEER values for SF/NAC NPs [38] and COS [18] groups was due to mechanisms for opening the tight junctions in the mucus layer of RPMI 2650 cells for this study. For the SF/NACNP group, NAC might inhibit the activity of protein tyrosine phosphatase (PTP) in the RPMI 2650 layer by forming di-sulfide bonds, so the junctions in the layer were opened up [38]. COS contains shorter N-glucosamine (N-Glc) units which might inhibit the activity of protein kinase C, so tight junctions are opened due to phosphorylation of tight-junction proteins [18,32]. To determine the dynamics of opening tight junctions for the two groups for 24 h, the TEER values were continuously monitored after NAC-loaded NPs or COS in the culture well were gently washed and refreshed by incubation media at 2 h. To determine the status of tight junctions in the cell layers, ZO-1 immuno-stains for the 300 μg/mL SF/NAC SF NPs and 0.5% COS groups were performed after 2 h of incubation with cells (Figure 5B). The micrographs for the ZO-1 (green) and the ZO-1/cell nucleus (or DAPI) merged stains in RPMI 2650 cell layers showed that there was more discreteness in ZO-1 (or green) stains in the layers for the NPs and COS groups than for the control group (Figure 5B), so more tight junctions in the layer were opened (Figure 5B). There were also more discrete green stains in the cell layer for both ZO-1 and the merged micrographs for the COS group than for the NPs group, which was consistent with the significant decrease in the TEER value for the COS group at 2 h (Figure 5A).

To determine the effect of SF/NAC NPs and COS on the dynamics of tight junctions in cell layers, the cell layers were washed using the culture medium at 2 h and the TEER values for three groups were continuously measured from 4 h to 24 h (Figure 5C). The TEER values for the control group for this period were similar to the values at 2 h. The TEER values increased to 90% of the value at 2 h and remained the same values from 4 to 24 h for the SF/NAC NPs group, but for the COS group, these values increased to different levels, depending on the concentration of COS that was incubated with the cells for 0–2 h, but these values were significantly different between the values at 2 h and for the 6–24 h period (e.g., about 62% of TEER values at 2 h for the 0.1% of COS case, n = 3, Figure 5C). There were different mechanisms for opening the tight junctions for SF/NAC NPs and COS, but this might be the first study to report different dynamic patterns in TEER values. The TEER values did not recover to early values at 2 h or tight junctions would be partially opened in the cell layers for the COS group, possibly because COS residues adhered strongly to the cell layer and were not washed away by the culture medium. Further study of this issue will be conducted.

### 3.6. Transcellular Transport of SF/NAC NPs: Uptake of NPs by RPMI 2650 Cells

Transcellular transport is an alternative pathway for nasal drug delivery. Therefore, the uptake of fluorescent FITC-SF and fluorescent SF/FITC-NAC NPs by RPMI 2650 cells was measured every 30 min for a 2 h period using a confocal microscope. The fluorescent images for cellular uptake studies were shown in Figure 6A.

The microscopic images in Figure 6A showed that the amount or intensity of green fluorescent cells for the uptake of SF/NAC NPs by RPM 2650 cells was significantly greater than the uptake for the SF NPs group from 30 to 120 min, respectively. For comparison, the value of the fluorescent intensity for the uptake of SF NPs by the cells in the first 30 min was assigned to 100% (Figure 6B). The difference in the intensity of the fluorescence uptake by cells for the two NPs for each 30 min was significant (*p* < 0.05 or better, n = 3) (Figure 6B). There was significantly higher fluorescent intensity for cellular uptake for the SF/NAC NPs group than for the SF NPs group (Figure 6B). The positive charges on SF/NAC NPs (Figure 1A) played an important role in increasing the uptake of the NPs by the cells, as shown in Figure 6A,B) [21]. These results were similar to those for the uptake of positively charged NPs in the Caco-2 cell layer for an intestinal epithelial cell model [21].

### 3.7. Cumulative Releases of NAC for NAC/SF NPs

The efficiency with which NAC was encapsulated in NAC/SF NPs is 62.9 ± 1.0% (n = 3) and the loading capacity for the NAC-loaded NPs is 12.6 ± 0.2%, as determined using Ellman’s method [28]. Table 1 shows the cumulative NAC release profiles for NAC in a buffer solution and NAC/SF NPs for 12 h. For a cumulative release of NAC for NAC solution (as a control), 70% and ~82% of NAC were released in a burst within 1 h and 2 h, respectively, and then NAC was released very slowly until 8 h (e.g., ~86.0%), which was the end of the release study. The cumulative releases of NAC for NAC/SF NPs are 58.8 ± 1.7% with a burst release for 4 h and then a slow release of NAC until 12 h (e.g., 61.4 ± 1.0 %, (n = 3)). The release of NAC for the NAC/SF NPs featured a slightly lower burst release and a NAC release from 2 h to 4 h or longer. (Table 1). Hence, the positively charged NAC/SF NPs could adhere in the nasal cavity to support NAC delivery for 4 h or longer and avoid fast mucociliary clearance of the nasal cavity (e.g., less than 30 min) [14]. However, the release period for NAC for NAC/SF NPs for this study was shorter than that for BSA or paclitaxel that was delivered by SF/polydopamine or SF NPs for wound healing or other applications [7,8]. Since SF/NAC NPs are designed as carriers for nasal delivery of NAC to the brain, dissolution mediums for studying the releases of NAC for the NPs in similar compositions of mucus are preferred and will be conducted in the near future.

### 3.8. In Vivo IVIS Study to Evaluate the Enhancement of Nasal Delivery of FITC-NAC to the Brain Using SF/FITC-NAC NPs

In vivo intranasal delivery of FITC-NAC, which was first produced according to the procedures stated in an early study with a few modifications [26], to rats, was measured using IVIS to determine whether SF/NAC NPs increased delivery of NAC to the brain regions and retention time of NAC in the nasal cavity than delivery using FITC-NAC solutions. Table 2 shows the intensity of the fluorescence or the amount of FITC-NAC that was delivered to the brain and the amount that was retained in the nasal cavity for intranasal delivery using fluorescent SF/NAC NPs or NAC solutions as carriers for 2 h. Those digitalized data were calculated using the fluorescent images of the IVIS study (e.g., parts of them are shown in Figure 7A,B).

The fluorescent intensity, or the amount of FITC-NAC that was delivered to the brain of rats using SF/FITC-NAC NPs as a carrier, was about 1.40 (e.g., 60 min) to 2.60 times (e.g., 15 min) more than that which was delivered by FITC-NAC solutions for five different sequences of measurements (Table 2). The amount of FITC-NAC that was retained in the nose of rats if SF/FITC-NAC NPs were used as a carrier was detectable between 1.26 ± 0.76 and 1.24 ± 0.61 (e.g., 30 and 90 min, respectively) (ph/s/cm^2^/sr, ×10^6^) for 30 min and 90 min, respectively. There was no difference between the baseline and zero for FITC-NAC that was carried by solutions because the fluorescent data had a large standard deviation. For instance, for t = 30 min and 120 min, the fluorescent intensity in the nasal cavity was 0.34 ± 0.39 (ph/s/cm^2^/sr, ×10^6^) and 0.75 ±1.19 (ph/s/cm^2^/sr, ×10^6^), respectively. Figure 7A,B showed the IVIS images for the fluorescent FITC-NAC delivered to the brain and retained in the nasal cavity of rats for using NAC solution and SF/FITC-NAC NPs as carriers between 30 and 90 min.

More FITC-NAC was delivered to the brain (e.g., an olfactory bulb in a brain, Figure 7B) and retained in nasal regions in rats if delivery used fluorescent SF/NAC NPs, as shown in Table 2. This confirmed the results of in vitro studies that showed enhancement of retention but excluded fast mucociliary clearance of the nasal cavity for SF/FITC-NAC NPs, which opened tight junctions in the RPMI 2650 cell layers if delivery was by fluorescent SF/NAC NPs and ensured fast RPMI 2650 cellular uptake of the NPs (Figure 5 and Figure 6). Interestingly, high intensity of the fluorescence in the left side of the brain was observed, which might be due to hardly controlling the exact intranasal locations of SF/FITC-NAC NPs to be injected into rats during the study. In addition, the fluorescent intensity for the nasal regions for FITC-NAC was also distributed at different locations (Figure 7B). Although the inherent structures of the nasal and brain of rats might play a role, it still needs to precisely inject those NPs in a future study.

## 4. Conclusions

SF/NAC NPs were produced by the desolvation technique, which was not cytotoxic to L929 cells at 6000 μg/mL (Figure 3). The high antioxidative activity was demonstrated by the scavenging capacity of the NPs for H_2_O_2_ and a MitoSOX stain for hBMSC (Figure 4A–C, respectively). The characteristics of opening tight junctions in RPMI 2650 cells by SF/NAC NPs were shown by decreasing TEER values and the discreteness in stains of ZO-1 proteins in cells (Figure 5A–C, respectively). The cellular uptake of SF/NAC NPs by RPMI 2650 cells was significantly more than the uptake of SF NPs when interaction times were increased (Figure 6A,B, respectively). The results for 2 h of the IVIS study for rats showed that the amounts of NAC delivered to the brain increased by 1.40–2.60 times and were retained longer in the nasal cavity by SF/NAC NPs than those did by NAC solutions (Table 1 and Table 2 and Figure 7A,B, respectively). In conclusion, multifunctional SF/NAC NPs were produced with high anti-oxidative activity and opened tight junctions in an RPMI 2650 cell layer, which enhanced the delivery of NAC to the brain of a rat, as shown by an IVIS study.

## Figures and Tables

**Figure 1 pharmaceutics-14-01288-f001:**
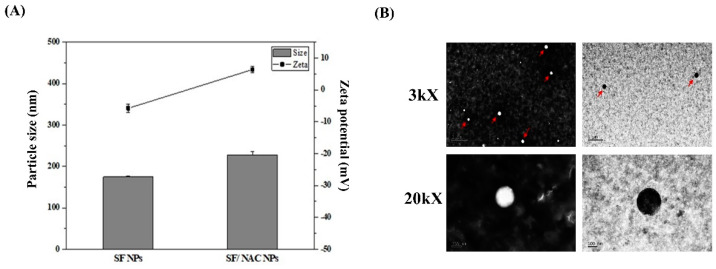
(**A**) Particle size and Zeta potential for SF and SF/NAC NPs. (**B**) The TEM morphology for SF and SF/NAC NPs. (bar: 1 μm and 100 nm for images of 3 kX and 200 kX, respectively).

**Figure 2 pharmaceutics-14-01288-f002:**
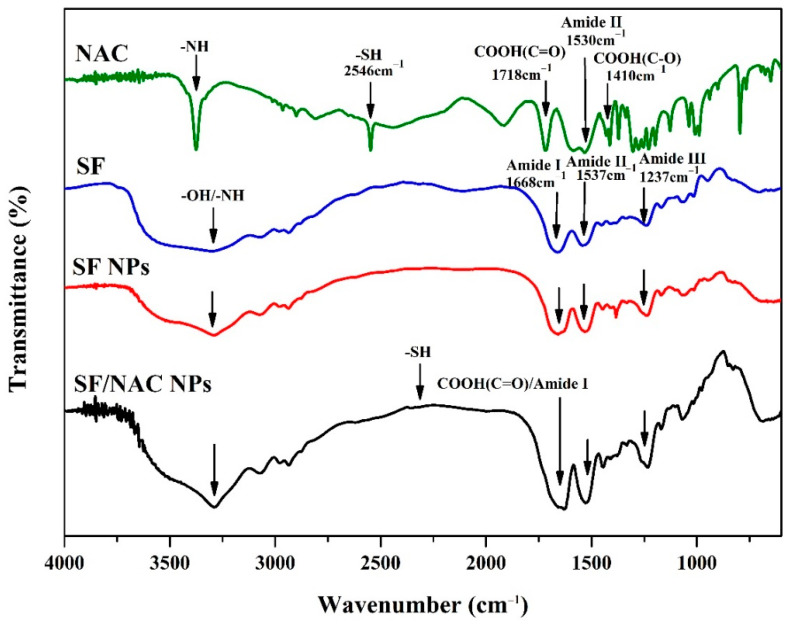
ATR-FTIR transmission spectra for analyzing functional groups for NAC, SF and SF/NAC NPs. The characteristic peaks for amide I, II and III (e.g., 1231 cm^−1^) in the SF spectra were also presented in the spectra for SF and SF/NAC NPs but the amide I peak for SF NPs was shifted from 1668 cm^−1^ to 1650 cm^−1^ for the spectra for SF/NAC NPs, possibly due to the chemical interaction between carboxyl groups (-COOH) in NAC and Amide I of SF.

**Figure 3 pharmaceutics-14-01288-f003:**
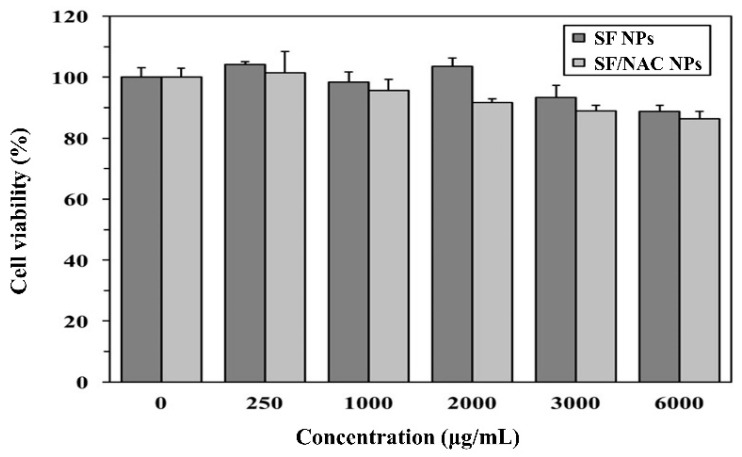
Cell viability studies for SF and SF/NAC NPs. The viability values (%) for the SF and SF/NAC NPs groups for concentrations of 0~6000 μg/mL were significantly (*p* < 0.01, n = 3, respectively) greater than 75%, the criteria for cytotoxicity evaluation of biomaterials of ISO10993-5. Hence, SF and SF/NAC NPs were biocompatible in these ranges.

**Figure 4 pharmaceutics-14-01288-f004:**
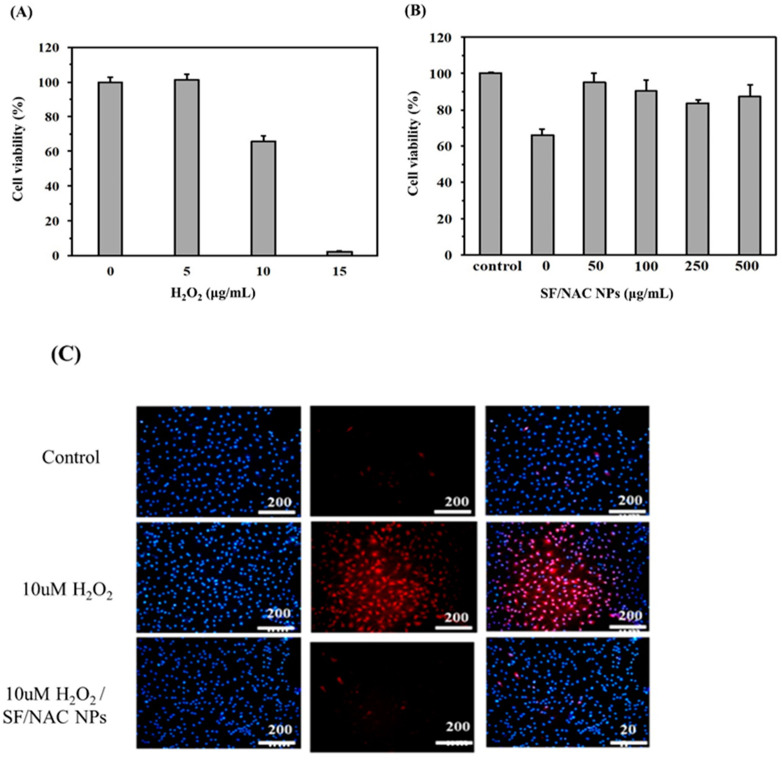
(**A**) HBMSC were incubated using different concentrations of H_2_O_2_ for 24 h to determine the amount of H_2_O_2_ that significantly reduced the cell viability of hBMSC. (**B**) Different concentrations of SF/NAC NPs were mixed with 10 μg/mL H_2_O_2_ and incubated with hBMSC to determine the viability of the cells and the anti-oxidative capabilities of SF/NAC NPs. (**C**) MitoSOX red stains for hBMSC cells were shown. The cells were incubated with 10 μg/mL H_2_O_2_ (red stain, MitoSOX) and 10 μg/mL H_2_O_2_/100 μg/mL SF/NAC NPs. There is no significant reduction in the intracellular ROS for the cells (rarely red stain) for the 10 μg/mL H_2_O_2_/100 μg/mL SF/NAC NPs group which was similar to the result for the control. (Note: stains for DAPI, blue; MitoSOX, red, respectively).

**Figure 5 pharmaceutics-14-01288-f005:**
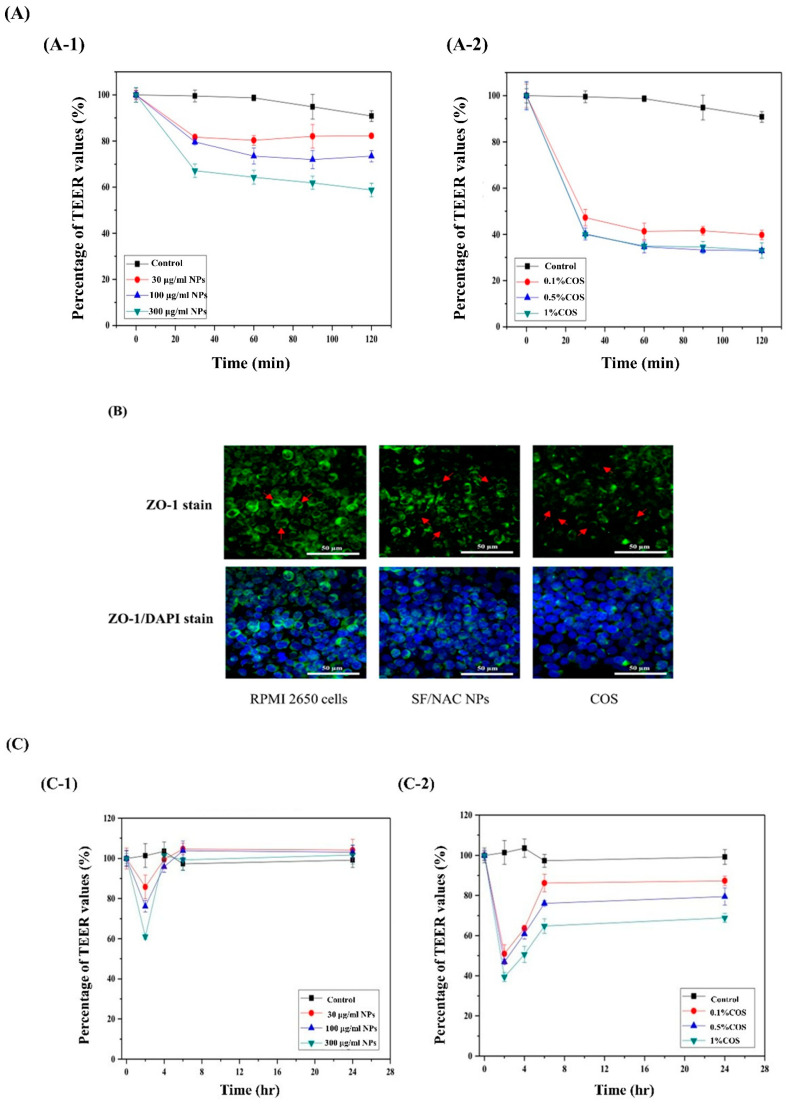
(**A**) The dynamics of TEER values for a RPMI 2650 cell layer that was incubated with various concentrations of (**A-1**) SF/NAC NPs and (**A-2**) COS for 2h (n = 3) at 30 min. There were fast decreases in the TEER values for both NPs and COS groups (e.g., at 30 min). This decrease was dependent on the concentrations of each group (n = 3), (**B**) Micrographs for ZO-1 stains (green cell circle) and ZO-1/cell nucleus (or DAPI) stains (green cell circle/blue) to determine the status of tight junctions in RPMI 2650 cell layers for three groups. There was greater discreteness (red arrows) in ZO-1 stains in the layer for SF/NAC NP and COS groups than for the control group so the more amounts of tight junctions in the layers for those two groups were opened, (**C**) The dynamic TEER values for RPMI 2650 cell layers were monitored and presented for 4 h to 24 h when (**C-1**) SF/NAC NPs and (**C-2**) COS containing medium were washed out after 2 h of incubation with the NPs and COS.

**Figure 6 pharmaceutics-14-01288-f006:**
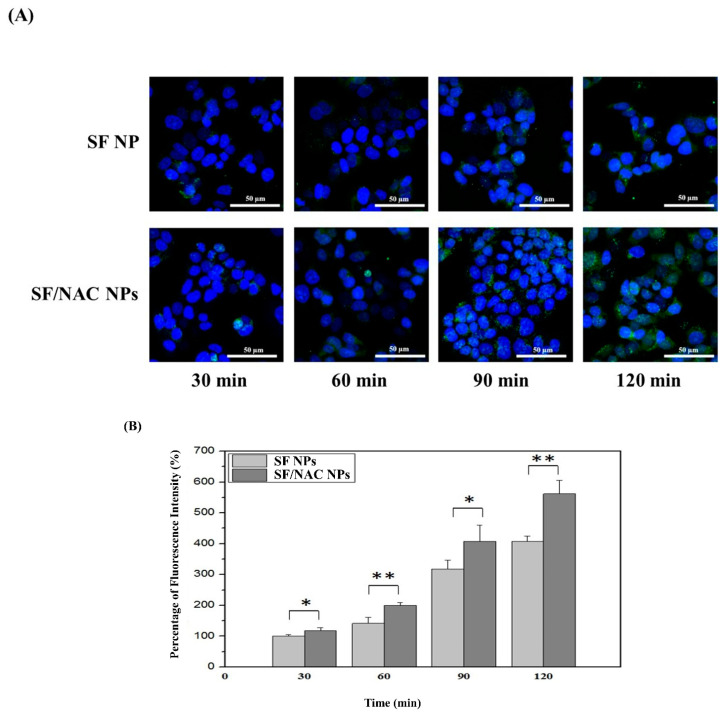
(**A**) Fluorescent images showing RPMI 2650 cellular uptake of fluorescent SF and SF/NAC NPs for a 2 h period, measured every 30 min. (**B**) The difference in the intensity of fluorescence for cells for the two NPs was significant (*: *p* < 0.05; **: *p* < 0.01, n = 4, respectively).

**Figure 7 pharmaceutics-14-01288-f007:**
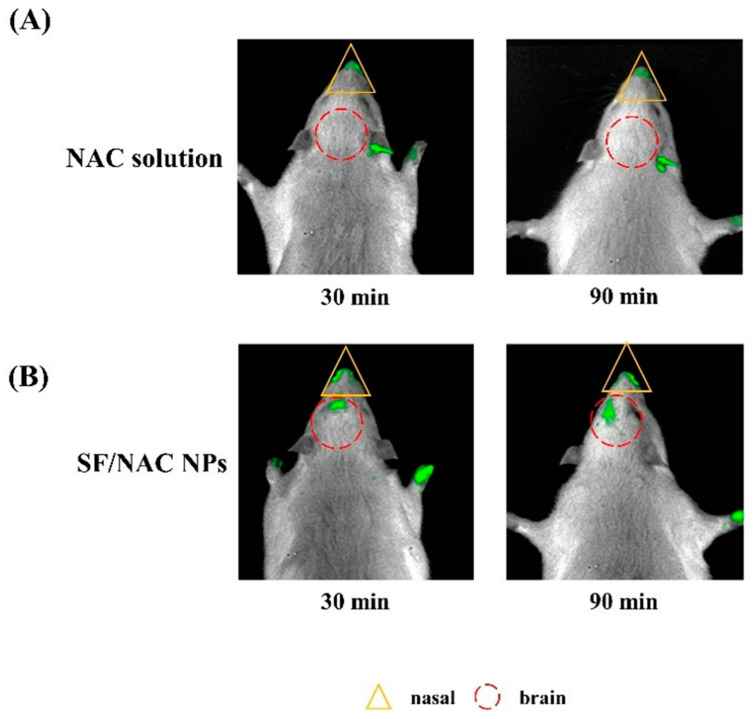
The IVIS images for the fluorescent FITC-NAC delivered to brains and retained in nasal cavities of rats for using a NAC solution (**A**) and SF/NAC NPs as carriers (**B**) at 30 and 90 min.

**Table 1 pharmaceutics-14-01288-t001:** The cumulative release of NAC for SF/NAC NPs (n = 4).

Time	0.5 h	1 h	2 h	4 h	8 h
**NAC (%)**	46.0 ± 0.1	69.6 ± 0.3	82.3 ± 0.7	84.7 ± 0.9	86.1 ± 1.4
**SF/NAC NP(%)**	32.6 ± 2.8	45.6 ± 3.7	55.3 ± 3.6	58.8 ± 1.7	60.3 ± 1.5

**Table 2 pharmaceutics-14-01288-t002:** The intensities of fluorescence of FITC-NAC that were delivered to the brains, and retained in the nasal cavities of rats for nasal delivery using SF/ FITC-NAC NPs as carriers or FITC-NAC solutions for 2 h were presented by an IVIS study. The fluorescent intensity in the nasal cavity for varying times was not different to that for t = 0 if delivery used NAC solutions (n = 4). All data were presented as mean ± sd., (n = 4).

FITC-NAC (ph/s/cm^2^/sr, ×10^6^)	15 min	30 min	60 min	90 min	120 min	Carries (n = 4)
**Brain region**	0.85 ± 0.15	1.18 ± 0.42	1.12 ± 0.36	1.10 ± 0.37	1.05 ± 0.40	NAC solution
**Brain region**	2.17 ± 0.73	2.14 ± 0.29	1.48 ± 0.13	1.72 ± 0.27	1.45 ± 0.12	SF/NAC NPs
**Enhancement Brain (100%)**	2.60 ± 1.05	1.96 ± 0.61	1.40 ± 0.39	1.68 ± 0.60	1.61 ± 0.91	SF/NAC NPs
**Nasal region**	0.81 ± 0.84	1.26 ± 0.72	1.05 ± 0.38	1.24 ± 0.61	1.04 ± 0.28	SF/NAC NPs

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
