# Peer review of "Antioxidative NAC-Loaded Silk Nanoparticles with Opening Mucosal Tight Junctions for Nasal Drug Delivery: An In Vitro and In Vivo Study"

_pharmaceutics, 2022, doi:10.3390/pharmaceutics14061288_

Round 1
Reviewer 1 Report
In this paper, the authors prepared NAC-loaded silk fibroin nanoparticles for nasal drug delivery. The article is interesting and can be published after minor revisions.
All the acronyms have to be defined when used for the first time. NP has been used in the first line of the abstract before its definition.
Some sentences in correspondence of lines 74-75 and 78-82 are highlighted. I think the grey color can be removed.
The authors said that the particles had been prepared by solvent dissipation. Please, add a few lines in the introduction on the technique.
Many techniques have been used to obtain nanoparticles, and I think they should be mentioned in the introduction. See, for example, supercritical carbon dioxide-based processes (10.1016/j.supflu.2010.06.001) or coacervation (10.1016/j.ijbiomac.2010.10.005).
There is too much white space before and after the figures. Please, remove it.
Author Response
For reviewer #1: Thanks for valuable comments.
Q1: All the acronyms have to be defined when used for the first time. NP has been used in the first line of the abstract before its definition.
A: We revised those acronyms (e.g., nanoparticles, NPs, the first line in the abstract).
Q2: Some sentences in correspondence of lines 74-75 and 78-82 are highlighted. I think the grey color can be removed.
A: We corrected them.
Q3: The authors said that the particles had been prepared by solvent dissipation. Please, add a few lines in the introduction on the technique.
A: We added several lines to introduced those techniques (P.6, paragraph No. 2, written in red words) and cited a review paper (Ref. No. 9).
Q4: There is too much white space before and after the figures. Please, remove it.
A: We carefully revised space problems.

Reviewer 2 Report
Comments: The authors have succeed to incorporate N-Acetyl-L-cysteine into silk fibroin nanoparticles. The work is very interesting . However, there are few issues, required some correction.
1. The cumulative drug release should have been studied in pH 6.5
2. The protocol of FITC conjugated SF/NACNPs was exactly missed
3. Scheme for chemical structure illustrates the possible attachment should have been provided.
4. The mechanism by which SF/NACNPs have ability to open tight junction was not explained.
5. Chitosan NPs can open tight junction, so they should have been used as reference in experiment
6. dissolution medium should have been contained mucus to produce real nasal environment
7. Authors have to explain why SF/NACNPs were seen in left side of rat?
8. The old REfs from 2001 to 2010 should to be updated
9. REf. 39 should to be corrected
10. It was written SF/NACNP in Fig (3,5 and 6). It should be corrected to SF/NACNPs
Author Response
For reviewer #2: Thanks for valuable comments.
Q1: The cumulative drug release should have been studied in pH 6.5
A: Please check the answer No.6.
Q2: The protocol of FITC conjugated SF/NACNPs was exactly missed.
A: We stated in Sec. 2.11, P.13. in red words. The methods to prepare FITC-NAC were stated. The FITC-NAC was then encapsulated into SF/FITC-NAC NPs by de-solvation technique using in this study.
Q3: Scheme for chemical structure illustrates the possible attachment should have been provided.
A: We stated the possible scheme for producing SF/NAC NPs in lines No. 1-8, Sec. 3.1, P.14. NAC was encapsulated into SF NPs which might be located at the hydrophilic segments of SF. The similar schematic diagram might refer to scheme 1, P. 12641, Ref. No.8 but the drug to be encapsulated and the locations of the drug in SF NPs were different from in NPs in this study.
Q4: The mechanism by which SF/NACNPs have ability to open tight junction was not explained.
A: The possible mechanisms were stated in lines no. 4-6, P.22 in red words with Ref. No. 38 for reference.
Q5. Chitosan NPs can open tight junction, so they should have been used as reference in experiment.
A: We stated the properties of COS solutions, which were used in this study, in Paragraph 2, P.20 and explained the possible mechanisms for opening tight junction in lines No. 6-8 in red words, P. 22 with Ref. No. 18 and 32 for references.
Q1&Q6: dissolution medium should have been contained mucus to produce real nasal environment
A: Two questions were about the same. We do agree to use a mucus analog solution as a dissolution medium to mimic real nasal environment to study NAC releases for the NPs. We used PBS as a dissolution medium for release study which might cause little difference in results compared to real nasal environment although drug-loaded SF NPs was not so pH sensitive according to our experiences. We noticed the reviewer suggestion, and will conduct those solutions for the future study. We wrote the red words in line No. 3-6, P.26.
Q7: Authors have to explain why SF/NAC NPs were seen in left side of rat?

Round 2
Reviewer 2 Report
Manuscript was being revised point by point according to reviewer comment.
However, few typos showed have to be fixed
1-μg/ml should to be replaced to μg/mL and all over the text.
2-"2.11. Rats studies " should to be changed to animal studies or In vivo studies
3- Few sentences are still in poor language state